# Treatment of De-Differentiated Liposarcoma in the Era of Immunotherapy

**DOI:** 10.3390/ijms24119571

**Published:** 2023-05-31

**Authors:** Maggie Y. Zhou, Nam Q. Bui, Gregory W. Charville, Kristen N. Ganjoo, Minggui Pan

**Affiliations:** 1Sarcoma Program, Division of Oncology, Department of Medicine, Stanford University School of Medicine, Stanford, CA 94304, USA; myzhou@stanford.edu (M.Y.Z.); nambui@stanford.edu (N.Q.B.); kganjoo@stanford.edu (K.N.G.); 2Department of Pathology, Stanford University School of Medicine, Stanford, CA 94304, USA; gwc@stanford.edu

**Keywords:** liposarcoma, immunotherapy, genomics

## Abstract

Well-differentiated/de-differentiated liposarcoma (WDLPS/DDLPS) is one of the most common histologic subtypes of soft tissue sarcoma (STS); however, treatment options remain limited. WDLPS and DDLPS both exhibit the characteristic amplification of chromosome region 12q13-15, which contains the genes *CDK4* and *MDM2*. DDLPS exhibits higher amplification ratios of these two and carries additional genomic lesions, including the amplification of chromosome region 1p32 and chromosome region 6q23, which may explain the more aggressive biology of DDLPS. WDLPS does not respond to systemic chemotherapy and is primarily managed with local therapy, including multiple resections and debulking procedures whenever clinically feasible. In contrast, DDLPS can respond to chemotherapy drugs and drug combinations, including doxorubicin (or doxorubicin in combination with ifosfamide), gemcitabine (or gemcitabine in combination with docetaxel), trabectedin, eribulin, and pazopanib. However, the response rate is generally low, and the response duration is usually short. This review highlights the clinical trials with developmental therapeutics that have been completed or are ongoing, including CDK4/6 inhibitors, MDM2 inhibitors, and immune checkpoint inhibitors. This review will also discuss the current landscape in assessing biomarkers for identifying tumors sensitive to immune checkpoint inhibitors.

## 1. Introduction

Liposarcoma (LPS) is the most common soft tissue sarcoma (STS) subtype in adults, accounting for up to 25% of all adult sarcomas [1,2,3]. Liposarcomas can be broadly categorized into three histologic subtypes: well-differentiated (WDLPS) and de-differentiated LPS (DDLPS), myxoid/round cell LPS, and pleomorphic LPS. WDLPS and DDLPS represent over 60% of all LPSs; however, treatment options remain limited [4]. A number of clinical trials targeting MDM2 are ongoing for patients with advanced de-differentiated liposarcoma. Here, we will review the clinical and genomic characteristics and novel therapeutic strategies involving immune checkpoint inhibitors and CDK4/6 and MDM2 inhibitors, and discuss several studies that explored potential biomarkers for predicting response to immune checkpoint inhibitors. 

## 2. Clinical Characteristics of WDLPS/DDLPS

WDLPS is primarily a loco-regional disease without metastatic potential. Local recurrences can occur in 30–50% of patients. In 10% of patients, especially those of retroperitoneum, mediastinum, and spermatic cord primary, WDLPS can de-differentiate into DDLPS [5,6]. WDLPS is sometimes also called an atypical lipomatous tumor (ALT) when they occur in the extremities and have a better prognosis [6,7]. 

DDLPS occurs primarily in adults over 40 years old, with equal gender distribution, and with retroperitoneum as the most common primary site. Extremities, paratesticular space, mediastinum, and head and neck are less common primary sites [6]. DDLPS accounts for more than 50% of all primary retroperitoneal sarcomas, and approximately 80–90% of them occur de novo [8,9]. The rest of DDLPS derives from the recurrence of WDLPS after an average of approximately 8 years from the onset of the diagnosis [6]. DDLPS can behave as an intermediate- and high-grade sarcoma, with approximately a 40% risk of local relapse and a 30% risk of distant metastasis. Biologically, it appears to be less aggressive compared to other pleomorphic sarcomas, such as undifferentiated pleomorphic sarcoma (UPS) [6,8,10]. 

Morphologically, WDLPS recapitulates the histologic characteristics of mature fat. Classically, WDLPS appears as lobulated, circumscribed, pale, soft, yellowish fat masses. The most predominant histological subtype of WDLPS is adipocytic (lipoma-like). Less common subtypes include sclerosing and inflammatory variants. Sclerosing WDLPS is the next most common WDLPS subtype and is frequently found in retroperitoneal and paratesticular sites. Sclerosing WDLPS shows scattered atypical stromal cells within the prominent collagenous stroma. Inflammatory WDLPS is a rare variant frequently found in the retroperitoneum and shows a chronic inflammatory infiltrate comprised of lymphocytes and plasma cells with lymphoid follicles.

DDLPS displays more heterogeneous morphologies with non-lipogenic components that are often characteristic of pleomorphic or unspecified spindle cell sarcoma [6,7,11,12,13]. DDLPS are often large, multinodular tumors that typically show moderate or high cellularity. Cytomorphology is variable, although significant nuclear pleomorphism is often observed. The stroma can be collagenous, myxocollagenous or myxoid, and even appear similar to high-grade myxofibrosarcoma. DDLPS can rarely also appear morphologically low-grade with bland, non-pleomorphic cytologic features, resembling desmoid fibromatosis. Despite their low-grade histological appearance, these DDLPS appear to have a similar ability as conventional DDLPS to metastasize.

WDLPS does not respond to systemic chemotherapy and is primarily managed with local therapy, including multiple resections and/or debulking procedures whenever clinically feasible [14]. Unlike WDLPS, DDLPS can respond to chemotherapy drugs and drug combinations, including Doxorubicin (or Doxorubicin in combination with ifosfamide), gemcitabine (or gemcitabine in combination with docetaxel), trabectedin, eribulin, and pazopanib [1,2]. However, the response rates are generally low with short duration [15,16,17].

## 3. Genomic Characteristics of WDLPS/DDLPS

WDLPS and DDLPS both exhibit the characteristic amplification of chromosome region 12q13-15. This region encompasses several genes, including *CDK4* and *MDM2*, *HMGA2* at 12q14.3, *CPM* at 12q15, *SAS/TSPAN31* at 12q14.1, and *YEATS4* at 12q15. *CDK4* (12q14.1) is located on a different amplicon from *MDM2* (12q15), which may explain why the two genes are not always co-expressed. *CDK4* is frequently, though not always, amplified in WDLPS/DDLPS [6]. *MDM2* amplification frequently includes *DDIT3,* which can sometimes cause confusion with myxoid liposarcoma [18,19]. *MDM2* is consistently amplified and overexpressed in WDLPS/DDLPS (nearly 100%) and is considered to be the main driver gene within the 12q amplicon. *HMGA2* is often coamplified with *MDM2* in WDLPS and DDLPS, but more commonly altered in benign lipomas [20,21,22,23,24]. *MDM2* amplification is not present in benign lipomas and is considered the molecular marker that distinguishes between benign lipomas and WDLPS [25,26]

DDLPS appears to exhibit higher amplification profiles of genes in the 12q13-15 region compared to that WDLPS [27]. There is evidence that DDLPS contained a higher ratio of *MDM2* amplification compared to WDLPS [28]. In addition, DDLPS contains additional genomic lesions, including the amplification of chromosome region 1p32, where the gene *JUN* is located, and chromosome region 6q23, where the activating kinase of *JUN* (*ASK1)* is located [29]. Amplifications of *JUN* and *ASK1* are thought to be mutually exclusive and found predominantly in DDLPS. Preclinical studies have shown that *JUN* amplification and overexpression block adipocytic differentiation in sarcomas, which may provide the pathologic basis of progression from WDLPS to DDLPS. These and other molecular lesions are likely responsible for the more aggressive behaviors of DDLPS compared to WDLPS [30,31]. A study showed that high *JUN* amplification (more than 16 copies) correlated with decreased DFS in DDLPS [32].

MDM2 is an E3 ubiquitin ligase that regulates the protein level and activities of p53 [33]. It acts as an oncogene by suppressing the p53 function, but in certain contexts, it has been found to act as a tumor suppressor [33,34]. In some retrospective studies, the degree of *MDM2* amplification in de-differentiated liposarcoma was found to be associated with poor overall survival (OS) and reduced interval from the time of resection to the time of recurrence [35,36]. *MDM2* amplification results in overexpression of MDM2 protein and its increased binding to p53, inhibiting its function and causing p53 degradation [33,34,37]. MDM2 can bind directly to the N-terminus of p53 and inhibit its transcriptional activation function. MDM2 prevents p53 from interacting with transcriptional co-activators and recruits transcriptional co-repressors to p53 (Figure 1). The C-terminal RING finger domain of MDM2 possesses E3 ubiquitin ligase activity that targets p53 for modification and subsequent degradation through the 26S proteasome [33,38]. p53 regulates *MDM2* expression by binding to its promoter, creating an autoregulatory feedback loop [33,39]. Several MDM2 inhibitors have been identified, and some of them have entered clinical trials [37,40].

CDK4, a cell cycle regulator, suppresses the retinoblastoma protein RB1 to stimulate cell cycle progression and the expression of a battery of genes, including *MDM2*. The *CDK4* gene encodes a 33 kD protein that forms molecular complexes with members of the cyclin D family, including cyclin D1, D2 and D3. The CDK4-CCND1 complex phosphorylates RB1 protein, which releases the E2F transcription factor, and in turn, up-regulates gene expression required for progression through the S-, G2-, and M-phases [41,42,43]. Amplification of *CDK4* and its cyclin partner *CCND1,* as well as the deletion of *CDKN2A* (an inhibitor of CDK4/6), is associated with worse survival in patients with advanced soft tissue sarcomas (Figure 2) [44,45]. A high copy number of *MDM2* and *CDK4* was shown to correlate with decreased disease-free survival (DFS) and disease-specific survival in DDLPS [32]. *TP53* mutations are also common in some soft tissue sarcomas and likely confer aggressive biological behaviors [45,46]. 

## 4. Novel Therapeutic Regimens in WDLPS/DDLPS

In Table 1, we have outlined some of the important clinical trials involving DDLPS that have been completed, including the trials that tested CDK4/6 inhibitors and immune checkpoint inhibitors. 

## 5. CDK4/6 Inhibitors

Three CDK4/6 inhibitors (abemaciclib, palbociclib and ribociclib) have shown clinical benefit in patients with advanced hormone receptor (HR)-positive/HER2-negative breast cancer when given in combination with endocrine therapy [60,61,62,63,64,65,66]. abemaciclib plus endocrine therapy has also demonstrated superior DFS over endocrine therapy alone in adjuvant settings in patients with high-risk early-stage breast cancer [63]. As monotherapy, only abemaciclib produced a meaningful overall response rate (ORR), while palbociclib and ribociclib resulted in primarily stable disease (SD) [60,67]. The ORR was 24.1% with abemaciclib at 150 mg twice daily and 32.5% with 200 mg twice daily in a phase II trial in patients with advanced breast cancer [68]. Despite the fact that CDK4/6 alterations are very common in many types of malignancies and that numerous preclinical studies on these agents have shown tumor suppressive effect, significant clinical benefits have only been demonstrated primarily in breast cancer and to a lesser extent, WDLPS/DDLPS [69,70,71]. Studies both in vitro and in vivo have shown that downregulation of CDK4/6 expression can inhibit the proliferation of liposarcoma cells by preventing the phosphorylation of Rb protein [72,73]. In addition, palbociclib can arrest soft tissue sarcoma cells in the G1 phase by inhibiting Wee1 kinase and also induce apoptosis and senescence [74,75,76]. 

In a phase I trial with palbociclib (PD 0332991), Schwartz et al. treated 33 patients with Rb-positive solid tumors or refractory non-Hodgkin’s lymphoma and found durable SD in 4 out of 7 patients with liposarcoma, subtype unspecified [47]. This led to the subsequent phase II trial by Dickson et al., which enrolled 30 patients with advanced WDLPS or DDLPS with *CDK4* amplification by fluorescence in situ hybridization and RB expression by immunohistochemistry (≥1+) [48]. The majority of these patients had retroperitoneal DDLPS [48]. The median progression-free survival (PFS) was 17.9 weeks, and the estimated 12-week progression-free survival (PFS) was 66%. One patient (3%) had a partial response (PR), and 30% had some degree of tumor shrinkage [48]. The dosing of palbociclib in this trial was 200 mg for 14 days in a 21-day cycle [48]. This trial was extended with an expansion cohort that enrolled an additional 30 patients (90% were DDLPS), and the dosing was changed to 125 mg for 21 days in a 28-day cycle which is the standard dosing for patients with advanced breast cancer [49,66,77]. The median PFS for this cohort of 60 patients was 18 weeks, and the 12-week PFS was 57%. There was one complete response (CR). Nine patients from the expansion cohort had paired tumor tissue biopsies performed, which revealed that downregulation of MDM2 expression mediated through ATRX was associated with clinical benefit [78]. Preliminary results from the TAPUR basket trial demonstrated the anti-tumor activity of palbociclib monotherapy in 29 patients with advanced STS with *CDK4* amplification and no *RB1* mutations (histologic subtype breakdown not reported). Patients were treated with palbociclib 125 mg for 21 days in a 28-day cycle. The median PFS was 16 weeks, and one patient had a PR [57]. Based on these data, National Comprehensive Cancer Network (NCCN) has recommended palbociclib as a treatment option for WDLPS and DDLPS [79].

A phase II study of abemaciclib 200 mg twice a day in 30 patients with DDLPS [53] reported a 12-week PFS of 76% and a median PFS of 30 weeks. In addition, three patients had more than 10% shrinkage of the tumor. These results appear favorable compared to the results from the palbociclib trials [48,49]. Based on these data, SARC041 is currently enrolling patients with advanced DDLPS for a placebo-controlled randomized trial to test the efficacy of abemaciclib in this patient population. Other trials that involved liposarcoma include a phase I trial with ribociclib in which six patients with liposarcoma had SD for more than 6 months; however, it was not clear if these six patients had WDLPS/DDLPS or other types of liposarcoma (pleomorphic or myxoid liposarcoma) [50]. 

The toxicities of palbociclib and abemaciclib in the phase II trials with WDLPS and DDLPS patients are similar to that of phase II and III trials in advanced breast cancer, including mild to moderate neutropenia and gastrointestinal toxicities [60,64,65,66,80,81,82]. Both agents are generally well tolerated. Abemaciclib appears to induce a lower degree of neutropenia compared to palbociclib [69].

A phase IB study testing the combination of CDK4/6 inhibitor ribociclib and MDM2 inhibitor Siremadlin in patients with WDLPS and patients with DDLPS showed limited activities, with 3 PRs out of 74 patients. In addition, there were ten patients who reached dose-limiting toxicities primarily associated with hematologic events [59]. This type of combination will require further exploration with different doses and particular types of molecules.

## 6. Immune Checkpoint Inhibitors

Immune checkpoint inhibitors (ICIs) have also been studied in WDLPS and DDLPS [83,84,85]. The SARC028 trial enrolled patients with advanced soft tissue and bone sarcoma who had progressed after at least one line of therapy with 86 patients enrolled and 80 patients evaluable, including ten patients each with UPS, DDLPS, synovial sarcoma and leiomyosarcoma, and 40 patients with osteosarcoma, Ewing sarcoma and de-differentiated chondrosarcoma [51]. Patients were given Pembrolizumab 200 mg intravenously every 3 weeks. The ORR for the entire cohort was 18%. Two out of 10 patients with DDLPS had a PR [51]. In the SARC028 expansion cohort [54], an additional 30 patients with UPS and 30 patients with DDLPS were enrolled for a total of 40 UPS and 40 LPS patients. The ORR for DDLPS was 10%, the median PFS was 2 months, and the 12-week PFS was 44%. In the Alliance trial A091401 [52], 96 patients with metastatic sarcoma were enrolled. Of the 85 eligible patients, 43 were randomized to single-agent Nivolumab and 42 to Nivolumab plus Ipilimumab. The ORR was 5% for Nivolumab and 16% for Nivolumab plus Ipilimumab. No response was seen in five patients with DDLPS. A meta-analysis including 27 trials with a total of 1012 patients treated with ICIs showed an 11% ORR for patients with liposarcoma and approximately 14% for all sarcomas [85]. Other meta-analyses and retrospective studies have shown similar ORR for advanced sarcoma [86,87,88,89]. 

The combination of Doxorubicin and Pembrolizumab has also been evaluated in phase I/II trials with 37 sarcoma patients. The primary endpoint was ORR by RECIST 1.1, with a 2-stage study design to rule out ORR of 15% or less with 85% power if the true ORR was 35%, using a 1-sided 5% level test. Unfortunately, accrual was closed at 31 of 35 planned patients because of an insufficient number of second-stage PRs, indicating that the study would not achieve the primary endpoint of ORR. ORR was 13% for phase 2 patients (19% overall). Two of 4 patients with DDLPS had durable PRs [55]. A similar single institution phase II trial with 30 patients treated with Doxorubicin and Pembrolizumab showed an ORR of 37% including 1 CR and 1 PR of 7 patients with liposarcoma [58]. The combination of Pembrolizumab with cryotherapy has also been studied in STS, including DDLPS [90]. A study using the combination of Pembrolizumab and Talimogene Laherparepvec in 20 patients with advanced sarcoma showed an ORR of 35% [56]. 

As illustrated above, CDK4/6 inhibitors and ICIs as monotherapy have demonstrated activity in DDLPS. However, it is not clear if the combination of these two classes of agents has synergistic activity. Several in vitro studies have shown that CDK4/6 inhibition modulates immune responses and may cooperate with ICIs to suppress tumor growth [91]. Preclinical studies have shown that abemaciclib activates the expression of endogenous retroviral elements in tumor cells and increases the intracellular level of double-stranded RNA, which stimulates the expression of type-III interferon and increased antigen presentation in the tumor microenvironment [92]. abemaciclib also increases tumor T lymphocyte infiltration, creating an inflamed T cell tumor microenvironment, and when combined with an ICI, leads to complete regression of tumors [93]. CDK4/6 inhibition by palbociclib or Trilaciclib enhances T cell activation by de-repressing NFAT signaling, leading to increased tumor T lymphocyte infiltration and activation of effector T cells. In addition, CDK4/6 inhibition up-regulates PD-L1 expression through suppression of the NF-kB pathway [94] and degradation of SPOP protein. CDK4/6 inhibition with anti-PD-1 immunotherapy enhances tumor regression and improves survival rate in a mouse model [95]. CDK4/6 inhibition suppresses CD4^+^ Tregs more than CD8^+^ cytotoxic T cells, increases the CD8^+^ T cells/Tregs ratio in the tumor microenvironment, and, when combined with an ICI, leads to the clearance of tumor cells by cytotoxic T cells [96]. These studies suggest the potential synergistic or enhanced effect of a CDK4/6 inhibitor and an ICI combination that could be evaluated prospectively in a clinical trial. 

## 7. ICI Biomarker Studies in Sarcoma

The search for biomarkers predictive of the benefit of ICIs has been an intense research effort [97]. Tumor mutation burden, PD-L1 expression of tumor cells, peripheral blood lymphocyte counts, and others have been associated with higher response rates and clinical benefits from ICIs [98,99,100,101]. However, in soft tissue and bone sarcomas, PD-L1 expression does not correlate well with a response or clinical benefit [86]. Several studies attempted to analyze the tumor microenvironment (TME) of sarcomas, including the composition of the tumor infiltrating CD4^+^ and CD8^+^ T cells, tumor-associated macrophages (TAM), and dendritic cells. Unfortunately, many of these studies were affected by a limited number of cases and histologic heterogeneity [102,103,104,105,106]. The complexity of TME is being increasingly revealed by ever more sophisticated techniques (e.g., single-cell RNA sequencing etc.) and well-designed studies in different malignancies.

The gene expression profiles of TME of 608 sarcoma specimens with a wide range of histology types were reported by Petitprez. In this analysis, five different phenotypes were described based on the immune infiltrates within the TME [107]. These five phenotypes included immune-low (A and B), immune-high (D and E), and highly vascularized (C). In addition, Group E showed a better response to Pembrolizumab and better survival compared to the other groups and was associated with the presence of tertiary lymphoid structures and enrichment of B cells within the tumors [107].

The immunostaining of 1072 sarcoma specimens showed that sarcomas with complex genomic lesions (mutations, copy number alterations) often contained a higher density of tumor infiltrating lymphocytes compared to fusion-driven sarcomas [108]. DDLPS and UPS were found to be the two top sarcomas that contained a higher density of tumor infiltrating lymphocytes among all the sarcomas [108]. In addition, this study showed that approximately 10–22% of all sarcomas had detectable PD-L1 or PD1 and that expression of PD-L1 and CD56 were associated with worse survival [108]. 

In a separate study, DDLPS was shown to contain higher density of CD68^+^ M1 macrophages and CD163^+^ M2 macrophages compared to fusion-driven sarcomas [109]. The study by Pollack et al. on 81 patients with sarcoma showed that tumor T-cell infiltration and clonality correlated with PD-1 and PD-L1 expression [110]. A phase II trial combining Pembrolizumab and metronomic Cyclophosphamide in 50 patients with advanced sarcoma showed little activity but found infiltration of M2 macrophages expressing indoleamine 2,3-dioxygenase (IDO) in the TME [111]. The phase I/II trial with Doxorubicin and Pembrolizumab showed that the presence of tumor infiltrating lymphocytes (TILs) is associated with worse OS in sarcoma [55]. Sarcomas can still respond to immunotherapy even without detectable PD-L1 expression [112].

The biomarker study of SARC028 tumor specimens pre- and post-treatment with Pembrolizumab suggested a correlation between the higher density of CD8^+^CD3^+^PD-L1^+^ activated T cells and response. Pre-treatment tumors with higher baseline density of effector memory CD8^+^ T cells and regulatory T cells (Tregs) showed a better response and better PFS [113]. These results are consistent with the results of the other studies [114]. It remains important and useful to investigate the mechanisms of response or the lack of response to ICIs in sarcoma through clinical trials as well as laboratory studies which could facilitate the development of novel therapeutics. 

## 8. Ongoing Clinical Trials

Several clinical trials for WDLPS and DDLPS are ongoing (Table 2). The most anticipated clinical trials for DDLPS currently are in the arena of targeting MDM2 with a newer generation of inhibitors. Nutlin-3a was the first specific small molecule MDM2 inhibitor developed that displaced MDM2 from p53 using its cis-imidazoline core structure [115,116,117,118]. The newer generation of MDM2 inhibitors has shown improved specificity and likely better activities in tumor inhibition [40,119]. Many of them are being tested for other diseases as well, including myelofibrosis and acute myeloid leukemia, etc. [120,121,122,123,124]. We will only discuss the clinical trials involving liposarcoma in this review. The “Treatment of Milademetan Versus Trabectedin in Patients With De-differentiated Liposarcoma (MANTRA)” study was just closed after completing enrollment. This trial enrolled 160 patients whose DDLPS had progressed on one or more lines of systemic therapy (at least one line of systemic therapy containing doxorubicin) and compared Milademetan versus Trabectedin with the primary endpoint of PFS. In the phase I trial, Milademetan produced a disease control rate of 58.5% and PFS of 7.2 months in the subgroup of patients with DDLPS in a phase I trial [125]. The clinical trial “Brightline-1: A Study to Compare BI 907828 With Doxorubicin in People With a Type of Cancer Called De-differentiated Liposarcoma” is testing another MDM2 inhibitor versus doxorubicin chemotherapy. This is a first-line trial that does not require previous exposure to chemotherapy. The trial plans to enroll 390 patients with PFS as the primary endpoint [126]. In a phase IA/IB dose escalation trial BI907828 appeared to show encouraging activities in patients with *MDM2*-amplified biliary tract cancer [127]. In preclinical studies, BI907828 showed excellent activities in de-differentiated liposarcoma xenografts carrying MDM2 amplification [128]. Some of the other MDM2 inhibitors have been studied in early-stage trials with limited activities, including SAR405838 and Siremadlin [129,130,131]. Another MDM2 inhibitor, ASTX295, is currently being tested in a phase I trial in patients with advanced de-differentiated liposarcoma. The clinical trial “Open-Label Study of the CDK4/6 Inhibitor SPH4336 in Subjects With Locally Advanced or Metastatic Liposarcomas” is testing a different CDK4/6 inhibitor for patients with WDLPS and DDLPS who have received no more than three previous lines of therapy with PFS as the primary endpoint. The clinical trial “palbociclib and INCMGA00012 (Retifanlimab) in People With Advanced Liposarcoma” testing the combination of palbociclib and a PD-1 inhibitor INCMGA00012 in patients with advanced WDLPS or DDLPS. Retifanlimab appears to have a similar toxicity profile with the other known anti-PD1 checkpoint inhibitor [132]. The clinical trial “ATX-101 in Advanced De-differentiated Liposarcoma and Leiomyosarcoma (ATX-101)” recruits patients with advanced DDLPS and LMS whose tumor had progressed on one line of therapy. ATX-101 is a small molecule peptide made of a novel human proliferating cell nuclear antigen (PCNA) interacting motif called APIM that is coupled to cellular and nuclear delivery domains and aims to target PCNA for cytotoxicity. In a phase I trial, no response was obtained, but 70% of patients (*n* = 20) had stable disease [133,134,135]. A dose escalation trial combining Vimseltinib and Avelumab is ongoing for advanced sarcomas, including DDLPS [136]. Vimseltinib is a CSF1R inhibitor initially developed for patients with recurrent tenosynovial giant cell tumor (TGCT) [137,138].

## 9. Conclusions

WDLPS and DDLPS are among the most common histological subtypes of STS. The standard systemic treatment options are limited to chemotherapeutic drugs. Recent data shows promising activity with CDK4/6 and immune checkpoint inhibitors in DDLPS. In addition, a number of clinical trials are evaluating the activity of small molecules targeting MDM2. Combination with a checkpoint inhibitor remains to be tested for safety and preliminary efficacy. Results from these trials shall be eagerly anticipated.

## Figures and Tables

**Figure 1 ijms-24-09571-f001:**
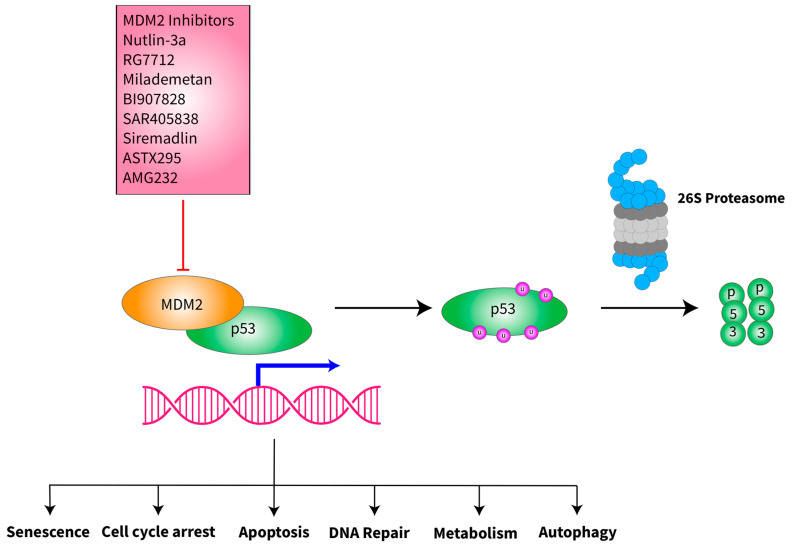
Schematic of MDM2 interaction with p53 and inhibition of MDM2 by its inhibitors. MDM2 is an E3 ubiquitin ligase that binds directly to the N-terminus of p53, bringing p53 to 26S proteasome for degradation, preventing the tumor suppressive function of p53, which regulates numerous cellular functions to safeguard the cells from oncogenesis. A number of MDM2 inhibitors have been synthesized and tested in vitro, in vivo and in early-stage clinical trials (see ongoing clinical trials section below). Many of these inhibitors bind to the MDM2-p53 interface and prevent the binding of MDM2 to p53, therefore stabilizing the protein levels of the wild-type p53 protein. The clinical trials with MDM2 inhibitors are therefore focused on malignancies that retain wild-type p53.

**Figure 2 ijms-24-09571-f002:**
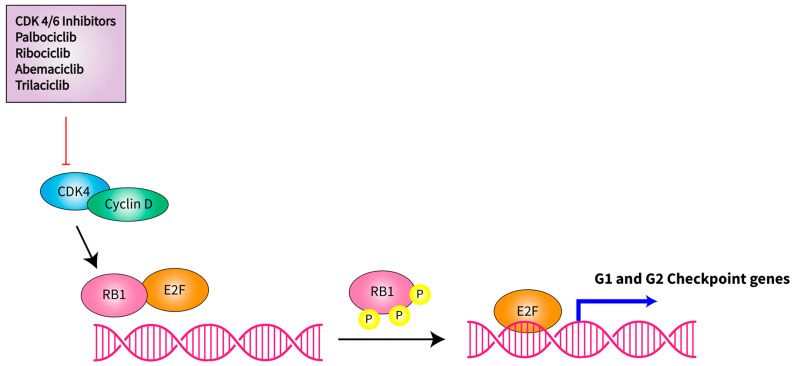
Schematic of CDK4 in cell cycle progression. CDK4 forms molecular complexes with members of the cyclin D family, including cyclin D1, D2 and D3 and becomes activated. The CDK4-CCND complex phosphorylates the retinoblastoma protein RB1 that binds to E2F to inhibit cell cycle progression. Phosphorylation of the Rb protein releases the E2F transcription factor, which subsequently stimulates the expression of genes that regulate the G1/S and G2/M checkpoint, leading to cell cycle progression.

**Table 1 ijms-24-09571-t001:** Select published trials.

Study	Year	Phase	Drug	Population	N	Response
Schwartz et al. [47]	2011	1	palbociclib	Rb-positive solid tumors or refractory NHL	33, including 7 LPS	SD in 4 out of 7 patients with LPS
Dickson et al. [48]	2013	2	palbociclib	WDLPS/DDLPS	30	1 PR, 66% PFS at 12 weeks, mPFS at 18 weeks
Demetri et al. [15]	2016	3	Trabectedin (T) vs. Dacarbazine (D)	LPS and LMS	54	LPS only: mOS 13.1 vs. 12.6 mo (T vs. D), *p* = 0.83
Dickson et al. (with expansion cohort of 2013 trial) [49]	2016	2	palbociclib	WDLPS/DDLPS	60	1 CR, 57% PFS at 12 weeks, mPFS at 17.9 weeks
Infante et al. [50]	2016	1	ribociclib	Rb-positive advanced solid tumors or lymphomas	132, including 39 LPS	SD for >6 months in 6 patients with LPS
Demetri et al. [16]	2017	3	Eribulin (E) vs. Dacarbazine (D)	LPS, excluding WDLPS	143	mOS: 15.6 vs. 8.4 mo (E vs. D)
Tawbi et al. (SARC028) [51]	2017	2	Pembrolizumab	STS and bone sarcoma	86, including 10 DDLPS	2 PR in DDLPS
D’Angelo et al. (Alliance A091401) [52]	2018	2	Nivolumab (N) +/− Ipilimumab (I)	STS	96 including 5 WDLPS/ DDLPS	No responses in WDLPS/DDLPS, total cohort: ORR = 5% vs. 16% (N vs. N + I)
Dickson et al. [53]	2019	2	abemaciclib	DDLPS	30	1 PR, 76% PFS at 12 weeks, mPFS at 30.4 weeks
Burgess et al. (SARC028 expansion cohorts) [54]	2019	2	Pembrolizumab	UPS and LPS	80, including 40 LPS	LPS only: 4 PR (ORR = 10%), mPFS 2 mo, mOS 13 months
Pollack et al. [55]	2020	1/2	Doxorubicin + Pembrolizumab	Anthracycline-naïve STS	37 including 4 DDLPS	2 of 4 patients with DDLPS had durable PRs, ORR = 19% overall
Kelly et al. [56]	2020	2	Talimogene Laherparepvec + Pembrolizumab	STS	20 (no LPS)	ORR = 35%
Schuetze et al. [57]	2021	2	palbociclib	STS with CDK4 amplification	20 (unknown LPS breakdown)	1 PR, mPFS 16.1 weeks, mOS 68.7 weeks
Livingston et al. [58]	2021	2	Doxorubicin + Pembrolizumab	Anthracycline-naïve STS	30, including 7 LPS	1 CR and 1 PR in liposarcoma, ORR = 37% overall
Razak et al. [59]	2022	1b	Siremadlin (p53-MDM2 inhibitor) + ribociclib	LPS	74	3 PR

Abbreviations: NHL = non-Hodgkin’s lymphoma, WDLPS = well-differentiated liposarcoma, DDLPS = de-differentiated liposarcoma, LPS = liposarcoma, LMS = leiomyosarcoma, STS = soft tissue sarcoma, UPS = undifferentiated pleomorphic sarcoma, PFS = progression-free survival, mPFS = median progression-free survival, PR = partial response, ORR = objective response rate, CR = complete response, mOS = median overall survival.

**Table 2 ijms-24-09571-t002:** Select ongoing trials in LPS.

Study	Drug	Study Design	Population	Primary Outcome
Brightline-1 (NCT05218499)	BI 907828 (MDM2 inhibitor) vs. Doxorubicin, first line	Phase 2/3	Advanced DDLPS	PFS
MANTRA (NCT04979442)	RAIN-32 vs. Trabectedin 1.5 mg/m^2^ every 3 weeks	Phase 3, open-label	Advanced DDLPS	PFS
SARC041 (NCT04967521)	abemaciclib 200 mg BID vs. Placebo	Phase 3, double-blind	Advanced DDLPS	PFS
NCT05580588	SPH4336 (CDK 4/6 inhibitor)	Phase 2, open-label with safety lead-in	Advanced LPS	PFS
NCT04438824	palbociclib + Retifanlimab (PD-1 mAb)	Phase 2, open-label with safety lead-in	Advanced LPS	Best ORR, RP2D
NCT03114527	ribociclib + Everolimus	Phase 2	Advanced DDLPS (arm A) and LMS (arm B)	PFR
NCT05116683	ATX-101 (small molecule peptide drug targeting proliferating cell nuclear antigen)	Phase 2, with safety lead-in	Advanced DDLPS and LMS	PFR
NCT04242238	Vimseltinib (CSF1R inhibitor) with Avelumab (anti-PDL1 antibody)	Phase 1b dose escalation and dose expansion	Advanced high-grade sarcoma including DDLPS	Best ORR, MTD

Abbreviations: DDLPS = de-differentiated liposarcoma, PFS = progression-free survival, BID = twice daily, LPS = liposarcoma, mAB = monoclonal antibody, ORR = overall response rate, R2PD = recommended phase two dose, LMS = leiomyosarcoma, PFR = progression-free rate, CSF1R = colony-stimulating factor-1 receptor, PDL1 = programmed death ligand 1, MTD = maximum tolerated dose.

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
