# Peer review of "Treatment of De-Differentiated Liposarcoma in the Era of Immunotherapy"

_ijms, 2023, doi:10.3390/ijms24119571_

Round 1

Reviewer 1 Report

Treatment of sarcoma still represents an unmeet need. Treatment progress in this field has been slow, and chemotherapy still represents the main therapeutic approach although responses in particular sarcoma types like liposarcoma are usually low and short.

The article brings into light new approaches in liposarcoma treatment field, including CDK4/6i and immunotherapy, and discuss new treatment data and main clinical trials.

The review is clear, and well written.

Author Response

Thank you for the positive comment, we are very appreciative.

Reviewer 2 Report

The manuscript is well-structured for publication.

The manuscript requires a minor degree of spelling/grammar check. 

Author Response

Thank you for the positive comment, we are very appreciative. We have also made changes with revision of some spelling/grammars.

Reviewer 3 Report

This is a great review and summarizes the actual literature very well. 

Author Response

(The authors gave the same response as above.)

Reviewer 4 Report

This paper Treatment of Dedifferentiated Liposarcoma in The Era of Immunotherapy highlights the clinical and genomic characteristics of WDLPS/DDLPS and novel therapeutic strategies, including CDK4/6 inhibitors and immune checkpoint inhibitors. The review demonstrates a contribution to the field and is in line with readers' interests of IJMS. However, there are still some shortcomings that need to be further improved or explained.

Comments:

Q1. There is too much background introduction in the abstract, and references are not recommended to appear in the abstract section.

Q2. Line 33, 4 A number of,what is the 4 mean? Line 34, a brief introduction of MDM2 is suggested to be supplemented, since it is firstly appeared here.

Q3. Do these ongoing clinical trials have presented some preliminary research results? I found that some references appeared in the 8. Ongoing clinical trials section.

Q4. If possible, I would suggest that some schematic representations of the anti-tumor mechanisms under different treatments (including CDK4/6 Inhibitors and Immune Checkpoint Inhibitors ) need to be supplemented.

Q5. Biomarkers are not a treatment strategies for cancer, the relevant presentations could be further improved.

Author Response

Thank you for the constructive criticism and suggestions, we have made changes accordingly.

Q1. There is too much background introduction in the abstract, and references are not recommended to appear in the abstract section.

Response: We have revised the abstract and removed some of the background information.

Q2. Line 33, “4 A number of”,what is the “4” mean? Line 34, a brief introduction of MDM2 is suggested to be supplemented, since it is firstly appeared here.

Response: We have cleaned up this. Thank you for pointing it out.

Q3. Do these ongoing clinical trials have presented some preliminary research results? I found that some references appeared in the ”8. Ongoing clinical trials” section.

Response: We have made good amount of changes including adding additional information and references about the clinical trials. Please see the revision in "Ongoing clinical trial" section.

Q4. If possible, I would suggest that some schematic representations of the anti-tumor mechanisms under different treatments (including CDK4/6 Inhibitors and Immune Checkpoint Inhibitors ) need to be supplemented.

Response: We have revised Figure 1 and 2 as suggested and included CDK4/6 inhibitors and MDM2 inhibitors respectively in each figure. Thank you for this excellent suggestion.

Q5. Biomarkers are not a treatment strategies for cancer, the relevant presentations could be further improved.

Response: We have revised this section as well accordingly.